# Parameter Localization of Greenhouse Gas Value Model and Greenhouse Gas Storage Simulation for Forest Ecosystems in China

**Mengdi Li [1,2], Yaoping Cui [1,2,3,*] , Yaochen Qin [1,2] , Oliva Gabriel Chubwa [1,2], Yiming Fu [1], Nan Li [1], Xiaoyan Liu [2] and Yadi Run [2]**

[1] Key Laboratory of Geospatial Technology for the Middle and Lower Yellow River Regions, Kaifeng 475004, China; lmd@cug.edu.cn (M.L.); qinyc@henu.edu.cn (Y.Q.); oliva2020@vip.henu.edu.cn (O.G.C.); fym0521@vip.henu.edu.cn (Y.F.); linan0716@henu.edu.cn (N.L.)

[2] College of Environment and Planning, Henan University, Kaifeng 475004, China; lxy@henu.edu.cn (X.L.); run@henu.edu.cn (Y.R.)

[3] Key Laboratory of Integrative Prevention of Air Pollution and Ecological Security of Henan Province, Kaifeng 475004, China

\* Correspondence: cuiyp@lreis.ac.cn; Tel.: +86-139-3785-8607

**Abstract:** Quantifying the greenhouse gas (GHG) storage in forest ecosystems can support global change directly, from a biogeochemical perspective. However, accurately assessing the amount of GHG storage in forest ecosystems still faces challenges in China because of their wide distribution, varying types, and the changing definitions and areas of forests. We used land-use data with 5-year intervals during 1990–2015 to investigate the spatiotemporal variations of forest ecosystems in China. As three major greenhouse gases in forest ecosystems, the potential storage of carbon dioxide, methane, and nitrous oxide can be calculated by a greenhouse gas value (GHGV) model. The results showed that the total area of forest ecosystems decreased by $15 \times 10^5$ ha during the study period. The area of forest ecosystems reached its highest level in 1995 and then declined. For various forest ecosystem types, shrubbery (Sh) increased by 0.82% but the broad-leaved forest, evergreen coniferous forest (ECF), and mixed forest (MF) all showed a downward trend. Correspondingly, the potential GHG storage of forest ecosystems declined from 156.97 Pg $CO_2$-equivalent ($CO_2$-eq) to 155.56 Pg $CO_2$-eq, a decrease of 1.41 Pg $CO_2$-eq. Compared with previous research results, the GHGV model proved to be an important supplementary method for estimating the potential storage of GHGs in forest ecosystems, especially in highly fragmented landscapes at a large scale. Our study indicated that the impact of forest ecosystems changes on potential GHG storage was serious during the study period. Our findings highlight that the GHGV model can be an effective and low-cost strategy to simulate the forest change and corresponding GHG storage. And considering the efficiency of the model and the historical analysis results of many periods, some of the results can also be used to inform the future afforestation programs and assess the expected GHG storage in China.

**Keywords:** forest; greenhouse gas; $CO_2$; $CH_4$; $N_2O$; greenhouse gas value model

## 1. Introduction

Practical strategies need to be developed to mitigate global warming, including increasing the ability of natural ecosystems to store atmospheric carbon [1,2]. Natural ecosystems regulate atmospheric $CO_2$ concentration because they sequester and release $CO_2$ at a far higher rate [3,4]. The effects of greenhouse gas (GHG) emissions in terrestrial ecosystems are the main drivers affecting GHG budgets [5,6].

Forest ecosystems are an important component in the climate system, and play a pivotal role in global biogeochemical cycling and climate regulation [7,8]. The carbon exchange between forests and the atmosphere is mainly through $CO_2$, which provides a reliable way to limit GHG emissions [9,10]. Analysis shows that 87% of the net $CO_2$ emissions between 1850 and 2000 were related to deforestation [11–13]. Plant biomass and soils are the potential carbon pools in forest [14,15]. Therefore, selecting effective measures to improve carbon storage in forest ecosystems has received considerable critical attention [16].

Forest soils are widely known as important terrestrial sinks for atmospheric GHGs [17]. The transformation of forest ecosystems also has an important impact on soil organic carbon storage [5,18,19]. However, previous research results on vegetation and soil organic carbon storage of forests have varied considerably [20]. The carbon storage of vegetation ranged from 4.38 Pg to 8.72 Pg [21–25], while the carbon storage of soil ranged from 17.39 Pg to 23.21 Pg [22,25,26]. Quantifying the major GHGs that are stored in forest ecosystems has become a central issue when assessing forest ecosystem services.

In recent decades, researchers have become interested in estimating carbon storage through machine learning and model simulation [27,28]. Studies have successfully calculated the carbon storage of terrestrial ecosystems by establishing a biomass model that converts biomass into carbon storage [29]. The greenhouse gas value (GHGV) model has been used to quantify the amount of GHGs released to the atmosphere if ecosystems were completely removed [30]. This model estimates the main GHGs from different ecosystems, including methane ($CH_4$) and nitrous oxide ($N_2O$) [31,32]. These studies provide new insights into spatial-temporal distribution of carbon storage in forest ecosystems through changes in GHG emissions [33]. However, most studies on GHGs are aimed at $CO_2$. The simulation of $CH_4$ and $N_2O$ storage in forest ecosystems at a regional scale is still relatively rare in China. Furthermore, the spatial distribution of different types of forest ecosystems may determine whether they are a carbon source or carbon sink. In summary, assessing the carbon storage in sub-type forest landscapes is indispensable for better evaluation of the ability of forests to store carbon [34].

In our study, we used the GHGV model to estimate the amount of $CO_2$, $CH_4$, and $N_2O$ stored in the forest ecosystems and compared the simulated results with localized parameters and original parameters. Furthermore, in order to verify the simulation results of our study, GHGV was converted to $CO_2$ by molecular weight and unit to compare the potential GHGs storage with existing research results.

## 2. Materials and Methods

### 2.1. Research Data

The data with 1 km spatial resolution during 1990–2015 was obtained from the Resource and Environmental Science Data Center of the Chinese Academy of Sciences (http://www.resdc.cn/) [35]. On the basis of China's vegetation map, we extracted evergreen coniferous forest (ECF), deciduous broad-leaved forest (DBLF), evergreen broad-leaved forest (EBLF), mixed forest (MF), and shrubbery (Sh) by converting the land-use data into the United States Geological Survey (USGS) ecosystem types.

According to the geographical characteristics of China [36], we divided China into seven eco-geographic regions: Northeast China (NE_Ch), Inner Mongolia (IM_Ch), Northwest China (NW_Ch), Qinghai-Tibet region (QT_Ch), Central China (C_Ch), East China (E_Ch), and South China (S_Ch). These classes helped to reveal the change characteristics of forest ecosystems in different regions.

### 2.2. Theory of Greenhouse Gas Value (GHGV) Model

The *GHGV* model for forest ecosystems estimates the continuous exchanges of *GHGs* that occur among forest ecosystems and the atmosphere over centuries by completely removing the ecosystems. Other disturbances that affected the *GHG* exchanges between the forest ecosystems and the atmosphere also were taken into consideration. The GHGV model converted other *GHGs* into their $CO_2$ equivalent to quantify the amount of GHG absorbed by forest ecosystems through radiative forcing. We set

the simulation time ($t_A$) as 100 years to indicate the biomass oxidation in forest ecosystems and the potential *GHGs* contribution to global warming. The *GHGV* model can be described as:

$$GHGV_{t_A}^{\delta_E} = \frac{\int_{t_A=0}^{\delta_A}\left[RF_{GHG}^{\delta_E}(t_A)w(t_A)\right]dt_A}{\int_{t_A=0}^{\delta_A}\left[RF_{pCO_2}(t_A)w(t_A)\right]dt_A},\tag{1}$$

where $RF_{GHG}^{\delta_E}(t_A)$ represents the enhanced radiative forcing as the GHG released after removing the specific ecosystem completely at $t_A$ during the $\delta_E$ period, and $RF_{pCO_2}(t_A)$ represents the radiative forcing value led by the CO$_2$ pulse at $t_A = 0$.

$$RF(t_A) = \sum_x a_x C_x^{\delta_E}(t_A),\tag{2}$$

where $a_x$ is the radiance of each GHG $x$ (i.e., CO$_2$, CH$_4$, N$_2$O), and $C_x^{\delta_E}$ is expressed by the following formula:

$$C_x^{\delta_E}(t_A) = \int_{t_E=0}^{\min(\delta_E,t_A)}\left[\frac{I_x(t_E)}{A}\rho_x(t_A - t_E)\right]dt_E,\tag{3}$$

where $I_x$ (kmol $\times$ ha$^{-1}$ year$^{-1}$) represents the amount of GHG $x$ transported from the forest ecosystems to the atmosphere; the molar value of the atmosphere expresses as $A$; and $\rho_x$ represents the current GHG in the atmosphere at $t_A$, which uses the attenuation of one pulse of GHG $x$ to estimate.

$$I_x(t_E) = S_x(t_E) - F_x(t_E)\tag{4}$$

as the organic matter to be cleared from the land (e.g., oxidative combustion and decomposition of organic matter), we express the potential released GHG as $S_x$, and $F_x$ is the annual GHG flux of the ecosystem.

When organic matter is decomposed and burned, the released GHG can be described as:

$$S_x(t_E) = \sum_p\left( OM_p\{ \begin{array}{cc} f_p^c E_{x,p}^c & t_E = 0 \\ (1 - f_p^c)E_{x,p}^d d_p(t_E) & t_E > 0 \end{array} \right)\tag{5}$$

where $OM_p$ (Mg dry matter/ha) is the organic biomass in the ecotype $p$-zone, $f_p^c$ and $1 - f_p^c$ are the proportions of the oxidative combustion and decomposition of organic matter, $E_{x,p}^c$ and $E_{x,p}^d$ (kmol $x$ Mg$^{-1}$ dry matter) represent the release ratio of the GHG $x$ produced by the oxidative combustion and decomposition for organic matter, $d_p(t_E)$ represents an exponential decay function, which indicates the annual decomposition rate of organic matter ($0 < d_p(t_E) < 1$).

### 2.3. Meta-Analysis of Localization Parameters

It is crucial to select suitable parameters for the GHGV model. The GHGV model has original global parameters for various ecosystems, but these needed further verification to see whether the model original parameters were suitable for China. To ensure more accurate simulation results, we collated data that related to localized GHGV parameters for forest ecosystem types at a large-scale in China from a literature search of relevant journal articles and carried out a meta-analysis. Based on the GHGV model and the availability of data that correspond to our research, vegetation biomass density (VBD), surface biomass density (SBD), underground root biomass density (URBD), litter/leaf organic matter density (LOMD), soil organic matter density (SOMD), and CO$_2$ flux (CO$_2$F) were considered.

For a broader representation of the parameters, we considered the parameters on the basis of large spatial and temporal scales, and we only took national research into consideration. We did not base our criteria on selecting long-term monitoring data. Although our simulation was over a long time (about 100 years), to ensure that the initial parameters were less sensitive to the model, we took average values if there were several referenced research results.

## 3. Results

### 3.1. Localized Model Parameters

Previous studies provided multiple data about the model parameters and we selected the averages for the model parameters after converting units for these values. [37]. The conversion coefficients of the forest between biomass and carbon were 0.5, which referred to the values reported by Fang et al. (2007) [22]. The carbon content of litter was based on Wang et al. (1999): humus was 0.58, trunk was 55.4, branches were 46.53, leaves were 45.84, and roots were 53.9 [38]. The VBD calculation was based on the vegetation resources inventory while the biomass amount was estimated using a biomass-accumulation model. The SBD and URBD was calculated from the vegetation biomass and rhizome ratio labeled by Huang et al. (2006) for different ecosystem types: forest was 0.265, and shrubbery was 0.91 [39]. The LOMD and SOMD data were collected from previous studies [24]. For the $CO_2F$, we counted the mean value of annual $CO_2F$ over many years (2000–2014) for forest ecosystems using the Carbon Tracker data products from the National Oceanic and Atmospheric Administration Earth System Research Laboratory [40]. Based on the above analysis, the GHGV model parameters were finalized as shown in Table 1.

**Table 1.** The greenhouse gas value (GHGV) model's localized parameters of China's forest ecosystems.

| Forest Ecosystem Type | VBD (Mg/ha) | SBD (Mg/ha) | URBD (Mg/ha) | LOMD (Mg/ha) | SOMD (Mg/ha) | $CO_2F$ (kmol/ha·Year) | Reference |
|---|---|---|---|---|---|---|---|
| Sh | 15.85 | 8.30 | 7.56 | 9.07 | 74.20 | 55.68 | |
| MF | 120.21 | 95.02 | 25.19 | 19.00 | 235.46 | 59.36 | |
| DBLF | 128.65 | 101.70 | 26.96 | 12.72 | 194.65 | 92.29 | [23,24,38,39] |
| EBLF | 222.59 | 175.96 | 46.63 | 10.63 | 187.09 | 37.00 | |
| ECF | 126.37 | 99.90 | 26.48 | 13.65 | 150.07 | 70.26 | |

Note: The acronyms in Table 1 are Vegetation Biomass Density (VBD), Surface Biomass Density (SBD), Underground Root Biomass Density (URBD), Litter/Leaf Organic Matter Density (LOMD), Soil Organic Matter Density (SOMD), $CO_2$ Flux ($CO_2F$); Shrubbery (Sh), Mixed forest (MF), Deciduous broad-leaved forest (DBLF), Evergreen broad-leaved forest (EBLF), Evergreen coniferous forest (ECF).

### 3.2. Spatial and Temporal Changes to Forest Ecosystems from 1990 to 2015

Figures 1 and 2 showed the spatio-temporal changes in China's forest ecosystems. Forest ecosystems were primary distributed in E_Ch, northern NE_Ch, and S_Ch, constituting approximately 74.7% of the total forest area in China (Figures 1 and 2). The forest ecosystems area in QT_Ch and C_Ch accounted for 21.9% of China 's total forest ecosystems, while the smallest area of forest (1.9%) was found in IM and NW_Ch.

Overall, from 1990 to 2015, the national forest ecosystems area decreased by $15 \times 10^5$ ha. There was an increase in area in 1995 and there was relative stability after 2000. There were minor fluctuations during the last 25 years in each region in China, but only NE_Ch decreased by 2.5%. The reduction of NE_Ch was mainly due to DBLF and MF. Of the various forest ecosystem types, MF covered the most area making up approximately 50% of the total forest, followed by Sh. Sh increased slightly (by 0.82%) from 1990 to 2015. The DBLF, EBLF, ECF, and MF areas decreased totally. In addition, DBLF showed the greatest decrease in area (4.46%).

### 3.3. Potential Greenhouse Gas (GHG) Storage in Forest Ecosystems in China

The total GHGs stored in China's forest ecosystems showed an overall-downward trend in our study period. The GHGV decreased by 1.41 Pg $CO_2$-eq during 1990–2015. Figure 3 showed an overview of the potential storage of GHGs in each region. The region with the largest GHGV was NE_Ch, and this region also showed the largest decline of 11.99 Pg $CO_2$-eq. Generally, the GHGV of regions other than NE_Ch in China remained relatively stable from 1990 to 2015. QT_Ch and S_Ch showed slight

decreases of 0.03 Pg CO$_2$-eq and 0.17 Pg CO$_2$-eq, respectively. IM_Ch, NW_Ch, and E_Ch increased slightly by up to 0.1 Pg CO$_2$-eq in the 25 years. For the different forest ecosystems (Figures 4 and 5), the GHGV of the DBLF, MF, and ECF decreased by 0.72 Pg CO$_2$-eq (4.46%), 0.68 Pg CO$_2$-eq (no more than 1%), and 0.12 Pg CO$_2$-eq, respectively. Only Sh increased, by 0.12 Pg CO$_2$-eq. The GHGs stored by DBLF and MF decreased by 0.42% and 0.33%, respectively, in NE_Ch. Only EBLF and Sh increased by 4.05% and 0.38%. The potential storage of GHG in forest ecosystems is critical to the GHG balance and climate regulation. The downward trend indicates that the capability of forest ecosystems to capture GHGs in China is declining.

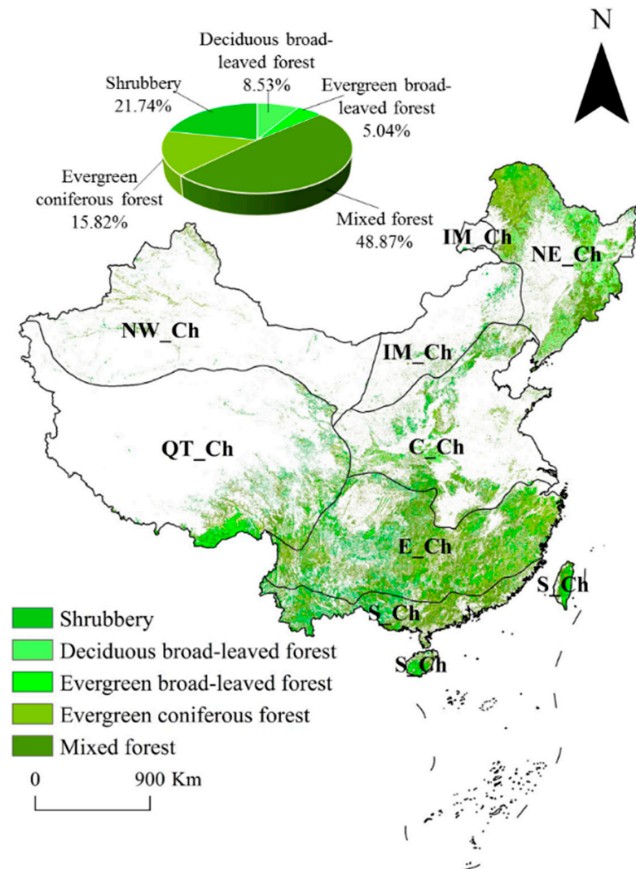

**Figure 1.** Spatial distribution and proportional area of forest ecosystems in China, 2015. The acronyms are Northeast China (NE_Ch), Inner Mongolia (IM_Ch), Northwest China (NW_Ch), Qinghai-Tibet (QT_Ch), Central China (C_Ch), East China (E_Ch), South China (S_Ch), and China, respectively.

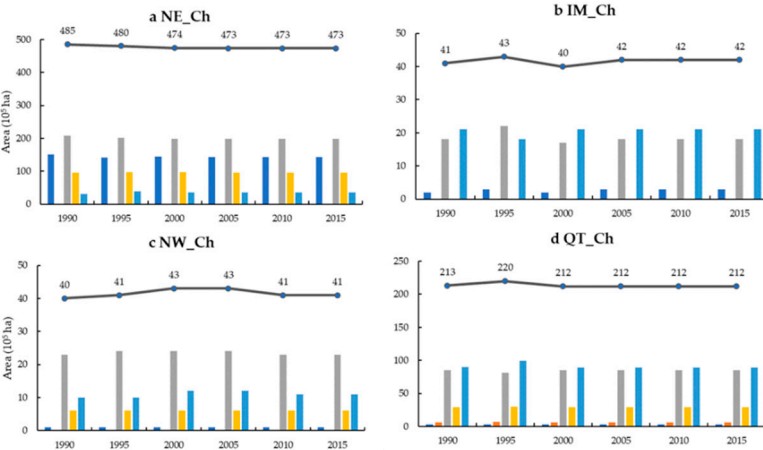

**Figure 2.** *Cont.*

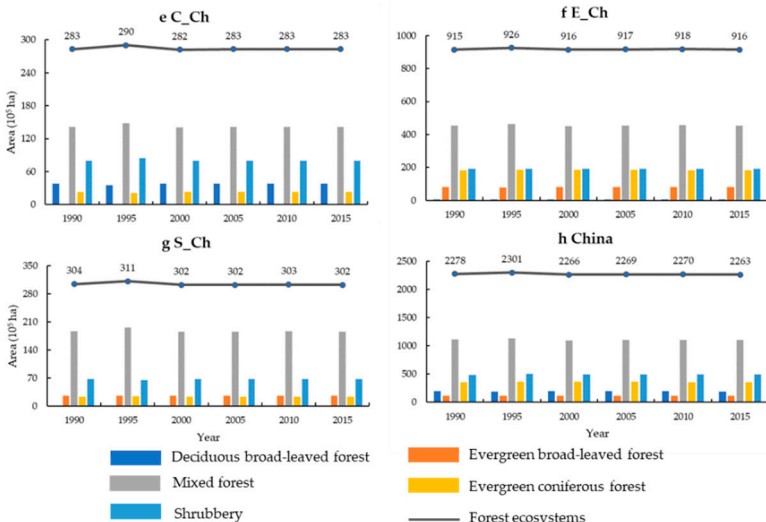

**Figure 2.** The area changes of China's forest ecosystems during 1990–2015. The acronyms in (**a**–**h**) are Northeast China (NE_Ch), Inner Mongolia (IM_Ch), Northwest China (NW_Ch), Qinghai-Tibet (QT_Ch), Central China (C_Ch), East China (E_Ch), South China (S_Ch), and China, respectively.

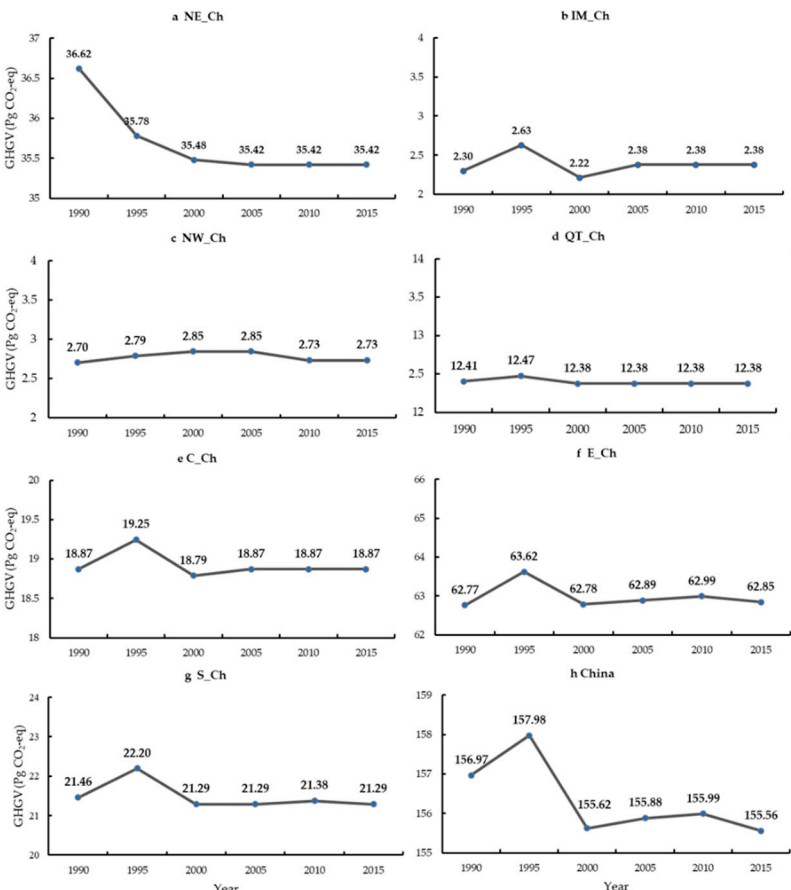

**Figure 3.** Changes of greenhouse gas value (GHGV) for forest ecosystems among various eco-geographical regions in China during 1990–2015. The acronyms in (**a**–**h**) are Northeast China (NE_Ch), Inner Mongolia (IM_Ch), Northwest China (NW_Ch), Qinghai-Tibet (QT_Ch), Central China (C_Ch), East China (E_Ch), South China (S_Ch), and China, respectively.

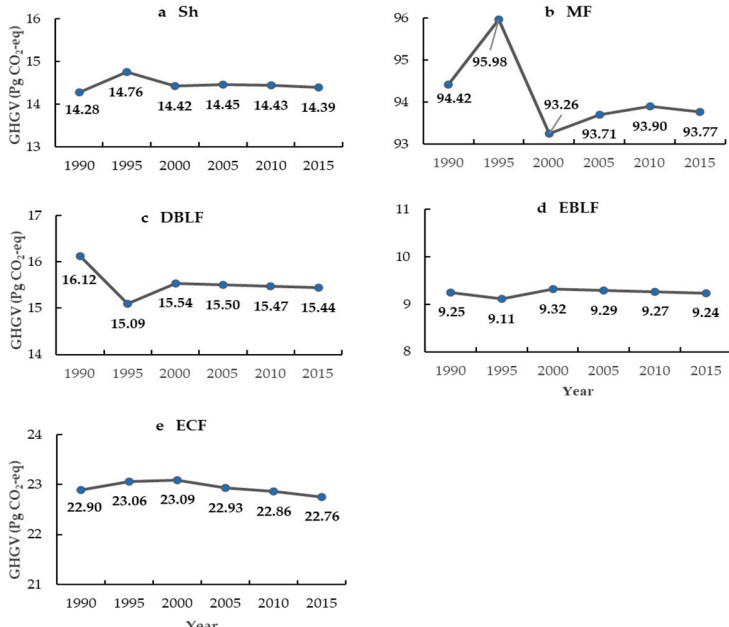

**Figure 4.** Changes of greenhouse gas value (GHGV) in China's various forest ecosystems during1990–2015. The acronyms in (**a**–**e**) are Shrubbery (Sh), Evergreen coniferous forest (ECF), Mixed forest (MF), Evergreen broad-leaved forest (EBLF), and Deciduous broad-leaved forest (DBLF).

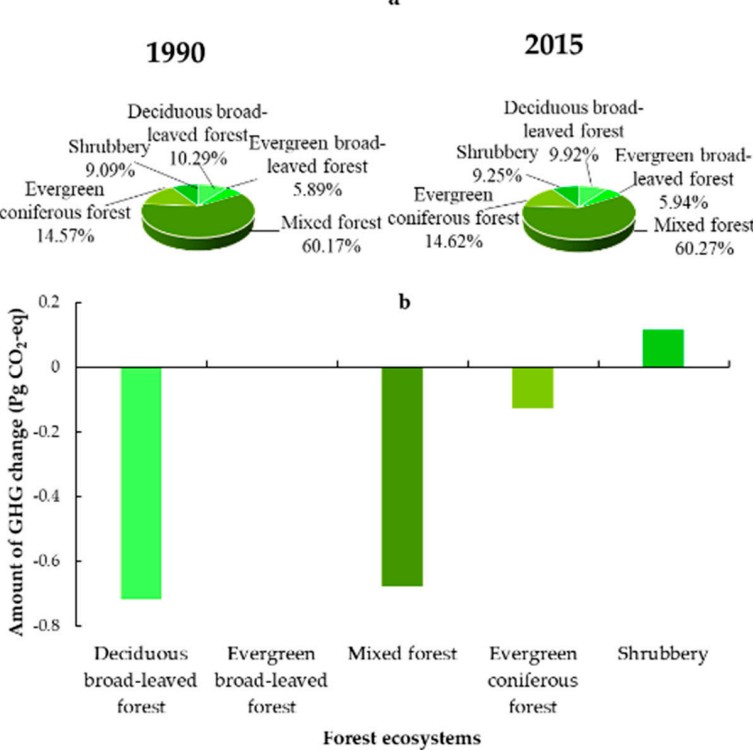

**Figure 5.** Greenhouse gas (GHG) change in different forest ecosystems: (**a**) Proportion of GHG in various forest ecosystems in 1990 and 2015; (**b**) Amount of GHG change in various forest ecosystems in China during 1990–2015.

*3.4. Comparison of Results with Localized Parameters and Model Original Parameters*

We also compared the GHGV using the original parameters (_Op) with that of the localized parameters (_Lp) for forest ecosystems in China. The comparison of the total GHGV for forest ecosystems with the two sets of parameters from 1990 to 2015 indicated that the model original parameters did not

perform well in China's forest ecosystems, with values far beyond those using localized parameters. The multi-year average GHGV of the model original parameters was higher than that of localized parameters; the difference between them was 120.3 Pg CO$_2$-eq. There were few inter-annual fluctuations in GHGV for localized parameters and model original parameters.

Figure 6 compares the results of localized parameters and those of model original parameters during 100 years. The GHGV in China's forest ecosystems increased fast in the first 10 years. The GHGV inter-year variations between BLF and MF in China were similar. The original global temperate forest remained far above that of other ecosystems during the simulation period. The GHGV differences between global coniferous forest and global Sh gradually reduced over 100 years. However, the GHGV annual change of DBLF, EBLF, and MF in the last 50 years increased by less than 1.5 Mg CO$_2$-eq/ha. The ECF GHGV in China was parallel with that of global coniferous forest. The GHGV of China's Sh was lower than that of global Sh during the simulation periods. However, although the amounts tended to vary, the GHGV trends in China's forest ecosystems were consistent with those of global forest ecosystems during the study periods.

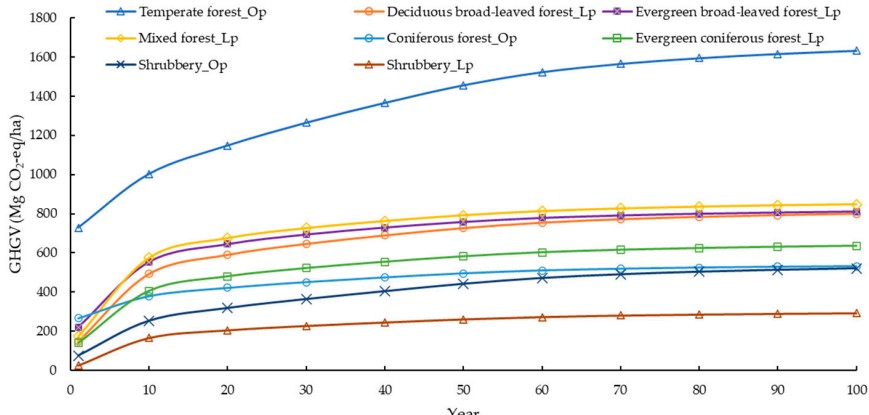

**Figure 6.** Temporal changes of greenhouse gas value (GHGV) in China's forest ecosystems (GHGV with localized parameters, _Lp) and global forest ecosystems (GHGV with model original parameters, _Op).

## 4. Discussion

Our comparison of the GHGV simulation results found large differences using the localized parameters and the model original parameters. We deliberated certain reasons for this. Our research developed the three forest ecosystem types of the original model into five forest ecosystem types. Although there were several studies on forest ecosystem types with detailed classifications that influenced the GHGV [41,42], the difference in forest ecosystem classification may have created heterogeneous uncertainty in the same forest ecosystem type.

Firstly, in our study, the main reduction in forest area was in NE_Ch. Secondly, the localized parameters and the original parameters of our research had some quite different values (Figure 7). Comparing the parameters between various forest ecosystems in China and globally, we found that the URBD of various forest ecosystems was similar to the SBD. The global temperate forest SBD was larger than that of BLF and MF in China. There were many secondary forests and young forests in China, which may cause the SBD values of the global coniferous forest and global Sh to be larger than corresponding forest types in China. Almost all the parameters varied from the original model parameters to a greater or lesser extent.

It can be hard to compare the potential GHGs storage with existing research results. In this study, the GHGV was converted to CO$_2$ by molecular weight and unit. Although there were many studies about carbon cycles in China's forest ecosystems, the useable research for our study were limited by sample data and analysis methods [23,43,44]. Table 2 provides the summary statistics for carbon storage and forest area from other studies and this study. The most obvious finding to emerge from the table was that the forest ecosystems area data we used was larger than others. The forest

ecosystems area ($227 \times 10^6$ ha) used in our study, were close to the eighth China forest resources inventory ($208 \times 10^6$ ha) and the interpreted area ($202 \times 10^6$ ha) based on multiple satellite data [25,45]. Although China's forest ecosystems can act as a carbon sink [46–48], we found that the carbon storage of forest ecosystems in China decreased by 0.152 Pg C/10 years, simulated by localized parameters. Deforestation and cultivated land reclamation may bring about a decrease in GHGs sequestered by forest ecosystems in most regions of China [49,50].

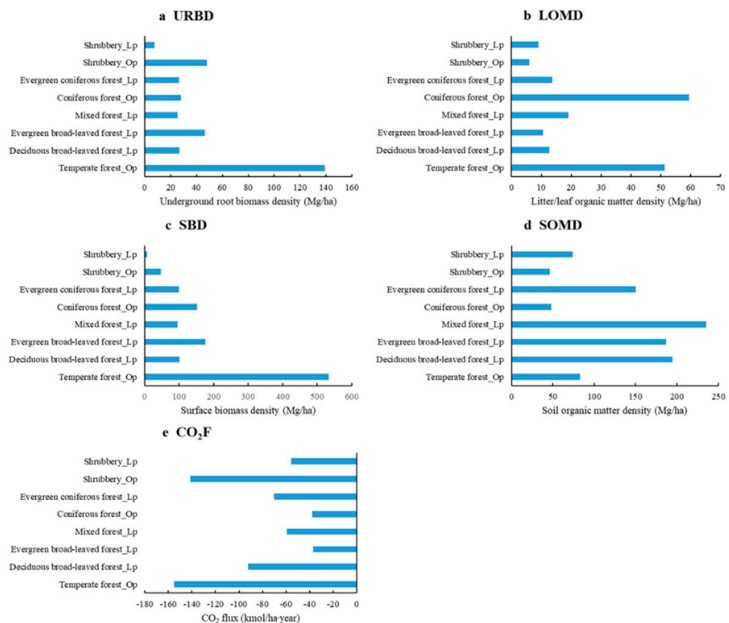

**Figure 7.** Comparison of model parameters for forest ecosystem types based on the localized parameters (their suffix is _Lp) and three similar ecosystem types with original global model parameters (their suffix is _Op) in China. The acronyms in (**a**–**e**) are Underground root biomass density (URBD), Litter/leaf organic matter density (LOMD), Surface biomass density (SBD), Soil organic matter density (SOMD), and $CO_2$ Flux ($CO_2$F).

**Table 2.** Results of carbon storage and forest area in previous researches and this study.

| Reference | Area ($10^6$ ha) | Carbon Storage (Pg C) | Carbon Density (Pg C/$10^6$ ha) |
|---|---|---|---|
| Fang et al. (1981) [21] | 116.5 | 16.38 | 0.14 |
| Zhou et al. (1989/1993) [24] | 108.62 | 28.11 | 0.26 |
| Li et al. (1981/1998) [23] | 121.63 | 31.93 | 0.26 |
| Zhao (1996/1999) [51] | 129.12 | 19.91 | 0.15 |
| Fang et al. (2000) [22] | 142.8 | 22.48 | 0.16 |
| The eighth forest inventory (2009/2013) [25] | 208 | 32.11 | 0.15 |
| Xie et al. (2004) [26] | 150 | 23.58 | 0.16 |
| This study (1990) | 227.8 | 42.81 | 0.19 |
| This study (2015) | 226.3 | 42.43 | 0.19 |

Our study also revealed that the GHGV model can support measured or inventory data. Although the analysis of actual observations is irreplaceable, it is undeniable that models are well suited to large scale simulations. Furthermore, the simulation results for different regions can be analyzed in a unified framework around the world. In terms of GHG density, our study was within the range of other studies. However, it is not useful to simply compare the GHG values, and more analysis should be conducted in combination with the area of the forest (Table 2). Another reason for the differences

among studies related to the centennial-scale simulation results are that they are not sensitive to variations over short time frames. Therefore, the limitations of the GHGV model itself still need to be considered in specific research.

## 5. Conclusions

This study analyzed the spatiotemporal changes in forest ecosystems in China from 1990 to 2015 and simulated the potential storage of GHGs in China's forest ecosystems using the GHGV model. We compared the model's original parameters and localized parameters. The area of China's forest ecosystems decreased by $15 \times 10^5$ ha between 1990 and 2015. Correspondingly, the simulation results indicated that the GHGs stored by China's forest ecosystems dropped from 156.97 Pg $CO_2$-eq to 155.56 Pg $CO_2$-eq.

Our study found that existing research results were a useful supplement to sampling observations. There were many significant advantages in model simulation on a large scale. The uncertainty in model factors driven by greater human disturbances and the differences among specific forest ecosystems at various stages of growth should be taken into consideration when using the GHGV model.

**Author Contributions:** Conceptualization, M.L. and Y.C.; methodology, M.L., Y.C., and Y.Q.; software, M.L., X.L., and Y.R.; formal analysis, Y.C., M.L., Y.F., N.L., and O.G.C.; writing-original draft preparation, Y.C. and M.L.; writing-review and editing, Y.C.; supervision, Y.C.; funding acquisition, Y.C. All authors have read and agreed to the published version of the manuscript.

**Funding:** This research was funded by Natural Science Foundation of China (42071415 and 41671425) and Outstanding Youth Foundation of Henan Natural Science Foundation (Y.C.).

**Acknowledgments:** The authors acknowledge the Resource and Environmental Science Data Center of the Chinese Academy of Sciences (http://www.resdc.cn/) for providing land use data.

**Conflicts of Interest:** The authors declare no conflict of interest.

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
