# Peer review of "Parameter Localization of Greenhouse Gas Value Model and Greenhouse Gas Storage Simulation for Forest Ecosystems in China"

_forests, doi:10.3390/f11111150_

Round 1

Reviewer 1 Report

Review of: "Parameter localization of greenhouse gas value model and greenhouse gas storage simulation for forest ecosystems in China"

This is a very simple book-keeping model of forest sector GHG balance.

It falls short of modern standards in forest sector GHG balance estimation and accounting in several respects.

There is no explicit account for effects of age class distribution, wildfire, harvest removals, and insect outbreaks on forests remaining forests.

It is not tracking the fate of organic matter in harvested products. No effects of management. There are no environmental change effects.

For these reasons, I have relatively low confidence in the estimates of change in GHG storage.

Reviewer 2 Report

Reference forests-907487:

Comments and Suggestions for Authors:

The work by Mengdi Li  et al. - Parameter localization of greenhouse gas value model and greenhouse gas storage simulation for forest ecosystems in China.  The research topic is of great interest and importance as the potential storage in forest ecosystems of three major greenhouse gases (carbon dioxide, methane, and nitrous oxide) during the 25 years.  However, due to the following main concerns, my suggestion is no acceptance for publication with this present version and major revisions are needed. 

Specific comments:

Page 2 (L85 to 87): We divided China into seven eco-geographic regions: northeast region (NEC), Inner Mongolia 85 (IM), Northwest China (NWC), Qinghai-Tibet region (TP), Central China (CC), East China (EC), and 86 South China (SC). - Have you divided the territory as you wish? Eco-geographic region can be divided as per one's wish?   

Page 6 (Figure 2): The figure should be self-explanatory but here those are obscure. It needs to be redrawn in a new way.

 Page 6 (L167 to 168): “The amount of GHGs sequestered by China’s forest ecosystems showed an overall-downward trend in our study period”. - This article discussed three types of gases (CO2, CH4 and N2O) but what is meant by sequestered?  CO2 and CH4 both are C base gases (CO2-C and CH4-C) which can be stored as carbon in the soil. Have you any evidence about N2O sequestration?  In general, when we discuss the combined effect of GHGs as radiative forcing, we transform all GHGs into CO2 equivalents. But you are talking about the sequestration of three gases in your manuscript.  

Figure 3 to 5: The graphs in this manuscript are not up to the mark.

.

 Page 10 (Figure 6): A series (with ten years interval) of up to fifty years but only two data point after that (70 and 100 year) which ruined the continuity of the chart.
